# Closed-Form Expressions of Upper Bound for Polarization-MDCSK System

**DOI:** 10.3390/e25091267

**Published:** 2023-08-27

**Authors:** Meiyuan Miao, Lin Wang, Weikai Xu

**Affiliations:** 1School of Communication and Information Engineering, Nanjing University of Posts and Telecommunications, Nanjing 210023, China; 2Department of Information and Communication Engineering, Xiamen University, Xiamen 361005, China; wanglin@xmu.edu.cn (L.W.); xweikai@xmu.edu.cn (W.X.)

**Keywords:** polarization modulation, *M*-ary differential chaos shift keying, closed expression

## Abstract

The performance analysis of polarization *M*-ary differential chaos shift keying (P-MDCSK) has been expressed using a tight upper bound with the Q-function. However, evaluating the Q-function directly is not a closed expression and there has been less work on closed expression for the upper bound. In order to solve the problem, this paper presents approximate closed-form expressions on the error probability of P-MDCSK. This expression is derived by employing a polynomial approximation of the Q-function. These closed-form expressions are verified through simulations conducted under both additive white Gaussian noise (AWGN) and multipath Rayleigh fading channels. The simulation results reveal that there exists only a negligible gap between the simulations and the derived closed-form expressions. For example, it is observed that the theoretical approximate closed-form expressions exhibit a marginal deviation of approximately 0.4 dB from the simulations when the bit error rate (BER) reaches $10^{-4}$. Although the proposed method can only give approximate closed-form expressions of the upper bound, it provides an effective method for other communication schemes where the exact BER closed-form formula cannot be obtained.

## 1. Introduction

Polarization modulation schemes have achieved much attention for their high spectral efficiency and low power consumption [1]; they transmit bits with polarization state on a plane constellation. The polarization shift keying (PolarSK) and corresponding schemes [2,3] are proposed and analyzed for high spectral efficiency. Another way to realize high spectral efficiency is dual-polarized modulation and 3D polarization modulation [4,5], although they suffer from polarization-dependent loss (PDL) on frequency selective fading channels [6,7]. M-ary differential chaos shift keying (MDCSK) with polarization has high spectral efficiency and without suffering from PDL [8]. The scheme utilizes the advantages of MDCSK, such as low cost, less complexity, little power, and excellent anti-interference capabilities over multi-path fading channels [9,10]. Moreover, it only requires a simple non-coherent demodulator without channel estimation and equalization [11].

In order to verify the correctness of the system performance, the bit error rate (BER) performance analysis is often carried out using the Gaussian approximation [12,13], which is feasible under AWGN and multipath environments. For the polarization modulation, the exact BER expression is difficult to obtain. Thus, the approximate BER expression is usually derived as a tight upper or lower bound [4,5]. For example, lower bounds have also been proposed in various scenarios [14,15]; they provide the corresponding lower bound-derived methods to different system environments and requirements. In [8], a unified upper bound of BER of polarization-MDCSK(P-MDCSK) was derived. However, for both the traditional MDCSK and P-MDCSK, their BER can be approximately expressed as an expression with a Q-function [12]. The Q-function is not a closed-form expression; in order to obtain the closed form of the BER expressions, a lot of work has proved that the closed form of BER can be courted [16,17,18] by representing the Q function approximately [19]. In this paper, we utilize polynomial approximation of the Q-function to obtain a simple closed-form BER formula for the P-MDCSK scheme.

The contributions of the paper are summarized as follows:

(1) The polarization *M*-ary differential chaos shift keying (P-MDCSK) is introduced, which uses the upper bound instead of the exact BER as a theoretical verification. Since the Q-function in the tight upper bound is not a closed form, the paper derives a close expression. It makes use of polynomial approximation instead of the Q-function.

(2) To achieve the closed expression, the bounds of the Q-function are used. It provides a generalized analytical expression of the closed-form on the upper bound, and calculates each of the three subsections in the upper bound for their different cases. The results show that there is only a very small gap between the simulations and the closed-form expressions

The study is organized as follows. Section 2 presents the system model of the P-MDCSK system briefly. Section 3 provides the closed-form expressions. Next, in Section 4, we show by numerical examples that the closed-form expression has a tight gap with simulations. Finally, Section 5 concludes the study.

## 2. System Model of P-MDCSK

A P-MDCSK constellation is characterized by a horizontal polarization state, a vertical polarization state, and a phase, which is shown in Figure 1. The information bit sequence consists of polarization and phase parts. By packing $l_{b}$ bits on the sphere and $n_{b}$ bits on the MDCSK phase, a total of $m_{c}=l_{b}+n_{b}$ bits can be conveyed. Thus, the $L_{b}=2^{l_{b}}$ symbols lie on the sphere and the $N_{b}=2^{n_{b}}$ symbol is in constellation with MDCSK, where $M=2^{m_{c}}$.

We assume that $L_{b}=2$ (This paper is based on the P-MDCSK in [8]). The $l_{b}$ is set to 1. Symbols lie on the sphere and $N_{b}$ symbols are with the MDCSK constellation. The transmitted signal with Stokes parameters [5] is written as
(1)$$\begin{array}{c} \begin{array}{c} S_{0}=E_{0}s_{x}\\ \;=\left[\begin{array}{c} E_{h}\\ E_{v} \end{array}\right]s_{x}\;=\left[\begin{array}{c} \cos\varphi e^{j\epsilon_{h}}\\ \sin\varphi e^{j\epsilon_{v}} \end{array}\right]s_{x}, \end{array} \end{array}$$
where φ is the angle of polarization, $\epsilon_{h}$ and $\epsilon_{v}$ are the phase of the signal in the horizontal and vertical state of polarization, respectively, considering $\vartheta =\epsilon_{h}-\epsilon_{v}=0$, φ=0,π/2, and $E_{h}^{2}+E_{v}^{2}=1$. The $s_{x}$ is the symbol with MDCSK modulation, which is written as
(2)$$\begin{array}{cc} \; & s_{x}=\left[s_{\mathrm{ref}},s_{\inf}\right]=\left[\underset{reference}{\underset{︸}{c_{x}}},\underset{information-bearing}{\underset{︸}{\cos\theta c_{x}+\sin\theta H\left(c_{x}\right)}}\right], \end{array}$$
where $c_{x}=\left[c_{x,1},c_{x,2},\ldots ,c_{x,i},\ldots ,c_{x,\beta }\right]$ is β-length chaotic signal. The H(.) denotes Hilbert transform operator; thus, $c_{y}=H\left(c_{x}\right)$. And θ is the phase of MDCSK modulation with θ∈[0,2π), where $\cos^{2}\theta +\sin^{2}\theta =1$.

The received signals of P-MDCSK over the multipath Rayleigh fading channel are expressed as
(3)$$\begin{array}{cc} \; & r_{h}=E_{h}s_{x}⊗h_{h,h}+E_{v}s_{x}⊗h_{h,v}+n_{h},\\ \; & r_{v}=E_{v}s_{x}⊗h_{v,v}+E_{h}s_{x}⊗h_{v,h}+n_{v}, \end{array}$$
where $r_{h}=\left[r_{h_{\mathrm{ref}}},r_{h_{\inf}}\right],r_{v}=\left[r_{v_{\mathrm{ref}}},r_{v_{\inf}}\right]$, are the received signals in the horizontal and vertical polarized states, and $h_{h,v}$ and $h_{v,h}$ are the composite gain of the input h/v and the output v/h polarization components. The $n_{h}$ and $n_{v}$ are the additive white Gaussian noise (AWGN) with zero mean and variance $N_{0}/2$, and ⊗ denotes the convolution operator.

The $h_{h,v}=h_{v,h}$ are set to 0 [8], and $h_{h,h}$ and $h_{v,v}$ have the same parameters as $h_{h,h}=h_{v,v}=\sum_{l=1}^{L}\alpha_{l}\delta \left(t-\tau_{l}\right)$, where *L* is the number of paths of the multipath channel, and $\alpha_{l}$ and $\tau_{l}$ are the channel coefficients and the path delay of the *l*th path.

The demodulation of the receiver is implemented in two parts: MDCSK and polarization states. Each part is demodulated by an independent process. In the polarization modulation part, considering the characteristics of differential modulation and polarization modulation, the maximum energy comparator is represented, where the *t*-th polarization state on the sphere is estimated by the following method:(4)$$\begin{array}{c} \hat{t}=\underset{t\in (h,v)}{\arg\max}(|ES_{h}|,|ES_{v}\left|\right), \end{array}$$
where $ES_{h}$ and $ES_{v}$ are expressed as
(5)$$\begin{array}{cc} \; & ES_{h}=r_{h_{\mathrm{ref}}}r_{h_{\inf}}^{T}+j\cdot H\left(r_{h_{\mathrm{ref}}}\right)r_{h_{\inf}}^{T},\\ \; & ES_{v}=r_{v_{\mathrm{ref}}}r_{v_{\inf}}^{T}+j\cdot H\left(r_{v_{\mathrm{ref}}}\right)r_{v_{\inf}}^{T}, \end{array}$$
where |·| denotes the absolute value, and H(·) is Hilbert transform operator. After $\hat{t}$ is determined, for the MDCSK part, the decision variables $z_{a}$ and $z_{b}$ are obtained as
(6)$$\begin{array}{cc} \; & z_{a}=r_{{\hat{t}}_{\mathrm{ref}}}r_{{\hat{t}}_{\inf}}^{T},\\ \; & z_{b}=H\left(r_{{\hat{t}}_{\mathrm{ref}}}\right)r_{{\hat{t}}_{\inf}}^{T}, \end{array}$$
where $\hat{t}$ is determined from *h* and *v*, $r_{{\hat{t}}_{\mathrm{ref}}}$ is either $r_{h_{\mathrm{ref}}}$ or $r_{v_{\mathrm{ref}}}$, and $r_{{\hat{l}}_{\inf}}$ is either $r_{h_{\inf}}$ or $r_{v_{\inf}}$, depending on $\hat{t}$. Then, the phase of MDCSK is decided by $z_{a}$ and $z_{b}$. The corresponding phase arccot$\left(z_{a}/z_{b}\right)$ and the decision boundaries are used for recovering the corresponding phase parts of information bits. It is important to remark that the MDCSK estimation depends on the estimation of $\hat{t}$. Note that the use of MDCSK does not affect the Stokes parameter.

The P-MDCSK detection algorithm is shown in Algorithm 1. The first step of the algorithm is polarization demodulation. After estimating the polarization state, $r_{{\hat{t}}_{\mathrm{ref}}}$ is selected from $r_{h_{\mathrm{ref}}}$ and $r_{v_{\mathrm{ref}}}$ and $r_{{\hat{t}}_{\inf}}$ is selected from $r_{h_{\inf}}$ and $r_{v_{\inf}}$. Then the corresponding phase arccot$\left(z_{a}/z_{b}\right)$ is estimated by MDCSK demodulation.
**Algorithm 1:** P-MDCSK detection algorithm**Input:** $r_{h_{\mathrm{ref}}}$, $r_{h_{\inf}}$, $r_{v_{\mathrm{ref}}}$, $r_{v_{\inf}}$**Output:** $b_{1}\ldots b_{l_{b}+n_{b}}$.
$D_{h}$←$r_{h_{\mathrm{ref}}}r_{h_{\inf}}^{T}$, $D_{v}$←$r_{v_{\mathrm{ref}}}r_{v_{\inf}}^{T}$$\hat{l}$←$\underset{l}{\arg\max}\left(\right|D_{h}\left|,\right|D_{v}\left|\right)$$z_{a}$←$r_{{\hat{l}}_{\mathrm{ref}}}r_{{\hat{l}}_{\inf}}^{T}$, $z_{b}$←$H\left(r_{{\hat{l}}_{\mathrm{ref}}}\right)r_{{\hat{l}}_{\inf}}^{T}$$\hat{\theta }$← arccot$\left(z_{a}/z_{b}\right)$$b_{1}\ldots b_{l_{b}+n_{b}}$←$\hat{l}$, $\hat{\theta }$

## 3. Closed Expression over Multipath Rayleigh Fading Channels

A tight upper bound is used to calculate the BER of polarization modulation [5]. Thus, in our previous work [8], we derived a BER upper bound of P-MDCSK, which is described as
(7)$$\begin{array}{c} P_{P-MDCSK}\leq P_{signal}+P_{polarization}+P_{joint}, \end{array}$$
where
(8)$$\begin{array}{c} P_{signal}=\frac{n_{b}}{l_{b}+n_{b}}P_{MDCSK}, \end{array}$$
(9)$$\begin{array}{cc} \; & P_{polarization}=\frac{1}{L_{b}\left(l_{b}+n_{b}\right)}\sum_{l_{s}=1}^{L_{b}}\sum_{l_{n}=1}^{L_{b}}D\left(l_{n}\to l_{s}\right)Q\left(\sqrt{\frac{d_{l_{n},l_{s}}^{2}}{2N_{0}}}\right), \end{array}$$
(10)$$\begin{array}{cc} \; & P_{joint}=\frac{1}{L_{b}N_{b}\left(l_{b}+n_{b}\right)}\sum_{l_{s}=1}^{L_{b}}\sum_{n_{s}=1}^{N_{b}}\sum_{l_{n}=1}^{L_{b}}\sum_{n_{n}=1}^{N_{b}}\left(D\left(l_{n}\to l_{s}\right)+D\left(n_{n}\to n_{s}\right)\right)Q\left(\sqrt{\frac{d_{l_{n},l_{s},n_{n},n_{s}}^{2}}{2N_{0}}}\right), \end{array}$$
where $D(l_{n}\to l_{s})$ is the Hamming distance, i.e., the number of different bits between symbols defined by $l_{n}$ and $l_{s}$, $l_{n}$ and $n_{n}$ denote the wrong symbols. The same definition is true of $D(n_{n}\to n_{s})$. The generic distance of $d_{l_{n},l_{s},n_{n},n_{s}}^{2}$ in the Euclidean space is expressed as
(11)$$\begin{array}{cc} \; & d_{l_{n},l_{s},n_{n},n_{s}}^{2}=2E_{k}\left(1-\left(\cos\left(\Delta \varepsilon -\frac{\Delta \vartheta }{2}\right)\cos\frac{\varphi_{l_{n}}}{2}\cos\frac{\varphi_{l_{s}}}{2}\right.\right.\;\left.\left.+\cos\left(\Delta \varepsilon +\frac{\Delta \vartheta }{2}\right)\sin\frac{\varphi_{l_{n}}}{2}\sin\frac{\varphi_{l_{s}}}{2}\right)\right), \end{array}$$
where $\Delta \varepsilon =\varepsilon_{n_{n}}-\varepsilon_{n_{s}}$, $\varepsilon_{n_{s}}$ is the MDCSK component of symbol *n*, $\Delta \vartheta =\vartheta_{l_{n}}-\vartheta_{l_{s}}$, $\left(\vartheta_{l_{s}},\varphi_{l_{s}}\right)$ is the polarization of the *l* symbol. The $d_{l_{n},l_{s}}^{2}$ is obtained from (Equation 11) when Δε=0 [8].

The closed expression is then calculated for each of these three components. The $P_{MDCSK}$ ($P_{DCSK}=\frac{1}{2}\mathrm{erfc}\left({\left[\frac{4}{\gamma_{s}}+\frac{2\beta }{\gamma_{s}^{2}}\right]}^{-\frac{1}{2}}\right)$, for $n_{b}$ = 1) for $n_{b}\geq$ 2 can be approximated to a simpler form in [12,20] as
(12)$$\begin{array}{c} P_{MDCSK}\approx \frac{2}{n_{b}}Q\left(\frac{\rho \pi }{2^{n_{b}}}\right), \end{array}$$
where $\rho =\sum_{l=1}^{L}\alpha_{l}^{2}2E_{c}/\delta =\sum_{l=1}^{L}\alpha_{l}^{2}E_{k}/\delta =2\gamma_{s}/\sqrt{2\gamma_{s}+\beta }$, and $E_{c}=E_{\hat{l}}^{2}\left(\sin^{2}\theta \sum_{i=1}^{\beta }c_{x,i}^{2}+\cos^{2}\theta \sum_{i=1}^{\beta }c_{y,i}^{2}\right)=\sum_{i=1}^{\beta }c_{x,i}^{2}=\sum_{i=1}^{\beta }c_{y,i}^{2}=\frac{E_{k}}{2}$, and $E_{k}=2E_{\hat{l}}^{2}\sum_{i=1}^{\beta }c_{x,i}^{2}=2\left(E_{h}^{2}+E_{v}^{2}\right)\sum_{i=1}^{\beta }c_{x,i}^{2}$ is the total energy of one symbol. The $\gamma_{s}=\sum_{l=1}^{L}\alpha_{l}^{2}E_{k}/N_{0}=\sum_{l=1}^{L}\alpha_{l}^{2}n_{b}E_{b}/N_{0}$ and $Q\left(x\right)=\frac{1}{\sqrt{2\pi }}\int_{x}^{\infty }e^{-\frac{t^{2}}{2}}dt$, for x≥0. Then, the total system BER of the proposed scheme over Rayleigh fading channel is given by (Here we base our performance analysis more on the system from a mathematical point of view in the original reference. In the real situation, it is necessary to consider whether there is a Rayleigh multipath fading in reality.)
(13)$$P_{multi}\approx \int_{0}^{\infty }P_{P-MDCSK}f\left(\gamma_{s}\right)d\gamma_{s}.$$
where $\gamma_{s}=\sum_{l=1}^{L}\alpha_{l}^{2}E_{k}/N_{0}$, and $f(\gamma_{s})$ is the PDF of $\gamma_{s}$ which can be found in [13].

In order to solve complex integrals of the BER expressions (Equation 13), the bounds of the Q-function are used. This part provides a generalized analytical expression of the closed form on approximate BER as the (Equation 7) is the upper bound of BER. However, this approach can still provide a trend of the BER. In the derivation of [19], a single-term exponential bound with adaptive parameters is considered. The general form of the bound is written as
(14)$$g\left(z\right)=\tau e^{-\theta z^{2}}\leq Q\left(z\right),\;for\;z\geq 0.$$
The function of g(z) is a lower bound if the τ and θ are established in [21] as
(15)$$\begin{array}{c} \theta \geq 0,\;and\;0<\tau \leq \sqrt{\frac{2e(\theta -1)}{\pi \theta^{2}}}. \end{array}$$
In order to simplify the calculation, the right side of the Formula (Equation 7) is expressed as
(16)$$P_{t}=P_{signal}+P_{polarization}+P_{joint},$$

Thus, (Equation 13) can be separated as
(17)$$\begin{array}{cc} \; & P_{multi}=\int_{0}^{\infty }P_{t}f\left(\gamma_{s}\right)d\gamma_{s},\\ \; & P_{ms}=\int_{0}^{\infty }P_{signal}f\left(\gamma_{s}\right)d\gamma_{s},\\ \; & P_{mp}=\int_{0}^{\infty }P_{polarization}f\left(\gamma_{s}\right)d\gamma_{s},\\ \; & P_{mj}=\int_{0}^{\infty }P_{joint}f\left(\gamma_{s}\right)d\gamma_{s}. \end{array}$$
By applying the bound to the given BER formula over the multipath Rayleigh fading channel, the lower bound of $P_{t}=P_{signal}+P_{polarization}+P_{joint}$ is calculated separately, and the lower bound of $P_{signal}$ becomes
(18)$$\begin{array}{c} {\mathrm{LB}}_{s}\left(\gamma_{s}\right)=F\int_{0}^{\infty }e^{-\theta \gamma_{s}}e^{-\gamma_{s}}d\gamma_{s}\leq P_{ms}, \end{array}$$
where *F* is expressed as
(19)$$\begin{array}{c} F=\frac{2}{n_{b}}\sqrt{\frac{2e(\theta -1)}{\pi \theta^{2}}}\sum_{l=1}^{L}\frac{\rho_{l}}{\gamma_{c}}\exp\left(-\frac{1}{{\bar{\gamma }}_{c}}\right)M_{s}, \end{array}$$
where $M_{s}=\frac{n_{b}}{l_{b}+n_{b}}$. Moreover, the integral in (Equation 18) is not analytically integrable. Thus, the exponential function is upper-bounded by [19]
(20)$$\begin{array}{cc} \; & e^{-z}\leq \frac{1}{1+z},\\ \; & e^{-\theta {\left(\frac{\rho \pi }{M}\right)}^{2}}=e^{-\theta \left(\frac{\pi^{2}}{M^{2}}\frac{4z^{2}}{2z+\beta }\right)}\\ \; & \leq \frac{M^{2}\left(2z+\beta \right)}{M^{2}\left(2z+\beta \right)+4z\theta \pi^{2}}, \end{array}$$
where $M=2^{n_{b}}$. The expression of the lower bound X can be derived by taking the sum of the items in (Equation 18) and replacing them with the upper bounds, which is expressed as
(21)$$X\left(z\right)=\frac{M^{2}\left(2z+\beta \right)}{\left(1+z\right)(M^{2}\left(2z+\beta \right)+4z\theta \pi^{2})}.$$
Thus, the integral of the closed expression of $P_{ms}$ over multipath Rayleigh fading channels can be written as
(22)$$\begin{array}{cc} \; & {\mathrm{LB}}_{s}\left(z\right)=F\int X\left(z\right)dz=M^{2}*\left[-\right.\\ \; & \frac{\left(M^{2}\left(2+\beta \right)-2M\beta -4\pi^{2}\beta \theta \right)\arctan h\left(\frac{M^{2}+4\pi^{2}z\theta }{\sqrt{M(M^{3}-4\pi^{2}\beta \theta )}}\right)}{\sqrt{M}\left(2M^{2}-M\beta -4\theta \pi^{2}\right)\sqrt{M^{3}-4\pi^{2}\beta \theta }}\\ \; & +\frac{\left(2-\beta )\ln(1+z\right)}{2M^{2}-M\beta -4\theta \pi^{2}}+\frac{\left(\beta -2\right)\ln\left(2M^{2}z+M\beta +4\theta z^{2}\pi^{2}\right)}{4M^{2}-2M\beta -8\theta \pi^{2}}]. \end{array}$$

Similarly, the closed expressions of $P_{mp}$ and $P_{mj}$ can also be derived by the same derivation process which is given in Appendix A. Finally, the total closed expression of $P_{t}$ can be expressed as LB(z)=LBs(z)+LBp(z)+LBj(z).

## 4. Numerical Results and Discussions

In this section, simulations are presented to verify the derived closed expression over multipath Rayleigh fading channel and AWGN. In all figures, SF denotes the spreading factor.

A three-path fading channel with equal channel average power gain is considered, i.e., $E\left[\alpha_{1}^{2}\right]=E\left[\alpha_{2}^{2}\right]=E\left[\alpha_{3}^{2}\right]=1/3$, and time delays $\tau_{1}=0,\tau_{2}=2T_{c},\tau_{3}=5T_{c}$. Here, $T_{c}$ is the sampling period of the chaotic signal $c_{x}$.

Figure 2 shows the comparisons between the derived closed expression of BER and the simulated BER of the P-MDCSK system over AWGN and multipath fading channels. The modulation order is M=8. The spreading factor is SF=64,128,256 over AWGN channel and SF=64,128,256,512 over multipath Rayleigh fading channels, respectively. The result shows that the closed expression is close to the simulations. The theoretical boundaries are roughly 0.4 dB away from the simulations when BER is $10^{-4}$.

The results of the simulation and the bounds analyzed in Section 3 for P-MDCSK are shown in Figure 3, with the parameters SF=128, θ=2, L=3 and M=4,8. The upper bound is very close to the simulated BER curves, and the trend and gap with the tight upper bound of the derived closed-form expression are in the reasonable range zone. The theoretical upper and the closed-form boundaries are 0.07 dB and 0.4 dB away from the simulations when BER is $10^{-4}$. The derived closed expression and upper bounds closely match the simulation curves and can constrain them from above and below. The simulation point at 23, 24, and 25 dB for 4-P-MDCSK is slightly higher than other points. It is caused by the theory $P_{MDCSK}$, whose theoretical derivation is an approximate expression, and the tendency of simulation and theory is similar in [12]. Thus, the BER performance of P-MDCSK over multipath Rayleigh fading channels can be predicted by upper and closed-form expressions.

## 5. Conclusions

The closed-form expressions on the tight upper bound for the P-MDCSK system are derived, in which the Q-function is approximated by polynomials. The derived closed-form expressions are expressed in three parts and they are verified by simulations over both additive white Gaussian noise (AWGN) and multipath Rayleigh fading channels. The results show that the theoretical boundaries are roughly 0.4 dB away from the simulations when BER is $10^{-4}$. Therefore, the closed expression not only reduces the computational complexity but also provides theoretical support for BER performance analysis for chaotic communication. The methodology can be used as a grounded method for other schemes in the same situation. This paper focuses on the mathematical optimization associated with a tight upper bound, and in the future, the method of obtaining a lower bound of P-MDCSK is a worthy focus for research.

## Figures and Tables

**Figure 1 entropy-25-01267-f001:**
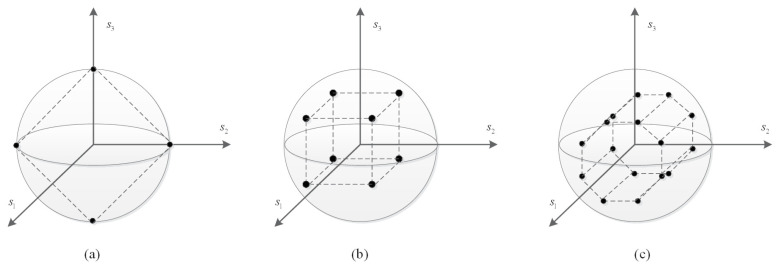
The polarization constellation of (**a**) *M* = 4, (**b**) *M* = 8, (**c**) *M* = 16.

**Figure 2 entropy-25-01267-f002:**
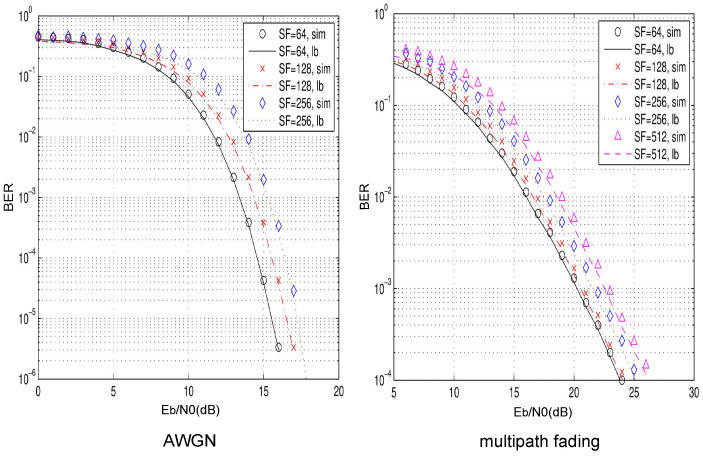
BER comparison between simulation (sim) and closed expression (lb) in P-MDCSK system over AWGN and multipath Rayleigh fading channels with M=8,SF=64,128,256.

**Figure 3 entropy-25-01267-f003:**
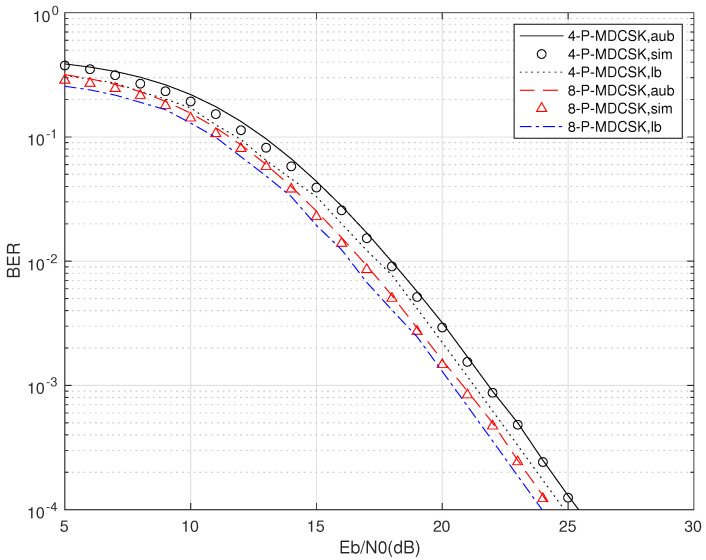
Simulation(sim), upper bound (aub) and closed expression (lb) performance comparisons of P-MDCSK multipath Rayleigh fading channels with SF=128, L=3.

## Data Availability

Not applicable.

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
