# Peer review of "Closed-Form Expressions of Upper Bound for Polarization-MDCSK System"

_entropy, 2023, doi:10.3390/e25091267_

Round 1
Reviewer 1 Report
- The article is written too concisely and assumes a close acquaintance with the topic.
- Without a preliminary reading of the papers [8, 15] it is difficult to understand the calculations given in the paper, since there is no sufficient description of the analyzed system and the notation used.
It is necessary to give a brief description of the P-MDCSK method, preferably with illustrations, in order to understand the symbols used in the paper. It is desirable to reveal the meaning of the symbols more fully.
- Line 58 meaning of the symbols rIinf, rIref is not explained.
- Page 3 line 10: the meaning of the symbols ln ls, nn, ns is not explained.
- Page 4 line 5: missing reference: [?]
- Page 4 line 6: in the article [15] θ>=0. Please, check the correctness of the formula.
- Formulas (15), (A1), (A9): What does fls means?
- What is dj in formula (15)?
Reviewer 2 Report
1. Abstract. The motivation can be further improved via adding the discussion of the practical application requirements for the developed methods.
2. Introduction. The contributions of this paper can be highlighted by points. Also, there is a lack of mathematical notations at the end of this section.
3. Introduction. The application of the approximate lower bound can be discussed. Also, the other related work can be added for a better understanding of this point. For example, the work in “Predictive precoder design for OTFS-enabled URLLC: A deep learning approach,” and "Approximate minimum BER power allocation for MIMO spatial multiplexing systems".
4. Section 2. The adopted results from the previous work can be briefly introduced to improve the readability of this paper.
Reviewer 3 Report
1. Overall, the paper appears to be incomplete in terms of technical writing. It seems that a more formal and technical refinement is required before undergoing re-evaluation.
2. I'm unsure of the intention behind obtaining an upper bound (4) and then subsequently deriving a lower bound for it. Is it perhaps an attempt to consider both as strong approximations? Furthermore, in the process of deriving intended results, bounds are being utilized without rigorous validation (to determine how tight they are). While the simulation results might coincidentally align, they cannot serve as a general validation for whether they genuinely correspond to close bounds.
3. Apart from simply utilizing a lower bound on the previously established Q function, there is essentially no substantial content. Mere application of existing lower bounds without introducing an original idea appears to fall short of the level of novelty required for publication in this journal. Furthermore, the derivation following the application of the lower bound does not appear to be particularly noteworthy either.
N/A
Round 2
Reviewer 3 Report
The authors have clearly addressed all my concerns.
Some minor typos exist:
ex 1) page2, line 38 and 46, etc
Thus, careful proof reading is suggested by the reviewer before publication.
N/A
